# Common intermediates and kinetics, but different energetics, in the assembly of SNARE proteins

Sylvain Zorman[1,2], Aleksander A Rebane[3,4], Lu Ma[1], Guangcan Yang[1], Matthew A Molski[1,2], Jeff Coleman[1,2], Frederic Pincet[1,2,5], James E Rothman[1,2]*, Yongli Zhang[1,2]*

[1]Department of Cell Biology, Yale University School of Medicine, New Haven, United States; [2]Nanobiology Institute, Yale University, West Haven, United States; [3]Department of Physics, Yale University, New Haven, United States; [4]Integrated Graduate Program in Physical and Engineering Biology, Yale University, New Haven, United States; [5]Laboratoire de Physique Statistique, UMR CNRS 8550 Associée aux Universités Paris 6 et Paris 7, Ecole Normale Supérieure, Paris, France

**Abstract** Soluble *N*-ethylmaleimide-sensitive factor attachment protein receptors (SNAREs) are evolutionarily conserved machines that couple their folding/assembly to membrane fusion. However, it is unclear how these processes are regulated and function. To determine these mechanisms, we characterized the folding energy and kinetics of four representative SNARE complexes at a single-molecule level using high-resolution optical tweezers. We found that all SNARE complexes assemble by the same step-wise zippering mechanism: slow N-terminal domain (NTD) association, a pause in a force-dependent half-zippered intermediate, and fast C-terminal domain (CTD) zippering. The energy release from CTD zippering differs for yeast (13 $k_BT$) and neuronal SNARE complexes (27 $k_BT$), and is concentrated at the C-terminal part of CTD zippering. Thus, SNARE complexes share a conserved zippering pathway and polarized energy release to efficiently drive membrane fusion, but generate different amounts of zippering energy to regulate fusion kinetics.

*For correspondence: james. rothman@yale.edu (JER); yongli. zhang@yale.edu (YZ)

**Competing interests:** The authors declare that no competing interests exist.

**Reviewing editor**: Axel T Brunger, Stanford University, United States

## Introduction

Soluble *N*-ethylmaleimide-sensitive factor attachment protein receptor (SNARE)-mediated membrane fusion is ubiquitous in eukaryotes and underlies numerous basic processes in humans, including neurotransmission, hormone secretion, and antibody production (*Sollner et al., 1993*; *Sudhof and Rothman, 2009*; *Wickner, 2010*; *Jahn and Fasshauer, 2012*). Malfunction of fusion has been associated with many important diseases such as neurological disorders and diabetes (*Burre et al., 2010*; *Stockli et al., 2011*). Consistent with their diverse functions and dysfunctions, these intracellular membrane fusion processes exhibit distinct kinetics and regulation (*Kasai et al., 2012*). For example, fusion of synaptic vesicles occurs within 0.2 ms in response to the arrival of an action potential (*Sabatini and Regehr, 1996*), whereas vacuole fusion in yeast is constitutive and lasts minutes (*Wickner, 2010*). Although these diverse processes have long been identified, it is not fully understood how SNAREs specialize in membrane fusion and become adapted to and regulated for various fusion speeds.

SNAREs constitute a large family of proteins with highly conserved modular structures (*Fasshauer et al., 1998*), including 38 SNARE proteins in humans. Each SNARE protein contains one or two defining SNARE motifs of around 60 amino acids in eight heptad repeats (*Figure 1A*). The motif is often connected to a C-terminal transmembrane domain via a short linker domain (LD of ~10 a.a.).

**eLife digest** Many processes in living things need molecules to be transported within, or between, cells. For example, damaged or waste molecules are transported within a cell to structures that can break the molecules down, while nerve impulses are transmitted from one neuron to the next via the release of signaling molecules.

Cells—and the compartments within cells—are surrounded by membranes that act as barriers to certain molecules. Vesicles are small, membrane-enclosed packages that are used to transport molecules between different membranes; and in order to release its cargo, a vesicle must fuse with its target membrane. To fuse like this, the forces that act to push membranes away from one another need to be overcome. Proteins called SNARES, which are embedded in both membranes, are the molecular engines that power the fusion process. Once the SNARE proteins from the vesicle and the target membrane bind, they assemble into a more compact complex that pulls the two membranes close together and allows fusion to take place.

The final shape of an assembled SNARE complex is essentially the same for all SNARE complexes; however, it is not known whether all of these complexes fold using the same method. Now Zorman et al. have used optical tweezers—an instrument that uses a highly focused laser beam to hold and manipulate microscopic objects—to observe the folding and unfolding of four different types of SNARE complex. All four SNARE complexes followed the same step-by-step process: the leading ends of the SNARE proteins slowly bound to each other; the process paused; then the rest of the proteins rapidly 'zippered' together.

Zorman et al. revealed that, although the steps in the processes were the same, the energy released in the last step was different when different complexes assembled. This suggests that the energy released by the 'zippering' of different SNARE proteins is optimized to match the required speed of different membrane fusion events. Furthermore, Zorman et al. propose that the reason why the majority of energy is released in the later stages of complex assembly is because this is when the repulsion between the two membranes is strongest.

The discoveries of Zorman et al. will now aid future efforts aimed at understanding better how the numerous other proteins that interact with SNARE proteins regulate the process of membrane fusion.

Complementary SNAREs are anchored to transport vesicles (v-SNAREs) and their targeted membranes (t-SNAREs) in disordered or partially disordered conformations. Their specific interactions lead to coupled folding and assembly into a stable parallel four-helix bundle, drawing the two membranes into close proximity for fusion (*Sollner et al., 1993*; *Sudhof and Rothman, 2009*; *Gao et al., 2012*). In the core of each SNARE bundle are 15 layers of hydrophobic amino acids and one middle layer of ionic amino acids. The ionic layer is formed by three glutamine residues (Q) and one arginine residue (R) from each of the SNARE motifs categorized as $Q_a$, $Q_b$, $Q_c$, and R SNAREs (*Fasshauer et al., 1998*; *Figure 1B*). Crystal structures show that the four-helix bundle structures are highly conserved in different SNARE complexes (*Sutton et al., 1998*; *Zwilling et al., 2007*; *Stein et al., 2009*), which can be aligned to the angstrom level (*Strop et al., 2008*).

The conserved sequences of SNAREs and their similar initial and final conformations implicate a conserved pathway of SNARE folding/assembly. However, the kinetics and energetics of SNARE folding have not been well characterized. It is notoriously difficult to study SNARE assembly using traditional ensemble-based experimental approaches due to the many states and pathways involved in the folding process, especially misassembled states (*Weninger et al., 2003*; *Pobbati et al., 2006*). In addition, functional SNARE assembly occurs in the presence of the opposing force imposed by membranes, which has a great impact on the kinetics and regulation of SNARE assembly (*Sudhof and Rothman, 2009*; *Gao et al., 2012*). Although studies of SNARE assembly are facilitated by the use of soluble SNAREs isolated from membranes, the lack of an essential force load may complicate data interpretation regarding functional SNARE assembly. For example, whereas complexin can suspend assembly of trans-SNAREs in a partially zippered state (*Kummel et al., 2011*; *Li et al., 2011*; *Malsam et al., 2012*), it cannot do so for isolated SNAREs (*Chen et al., 2002*). Thus, new methods are required to better elucidate SNARE assembly.

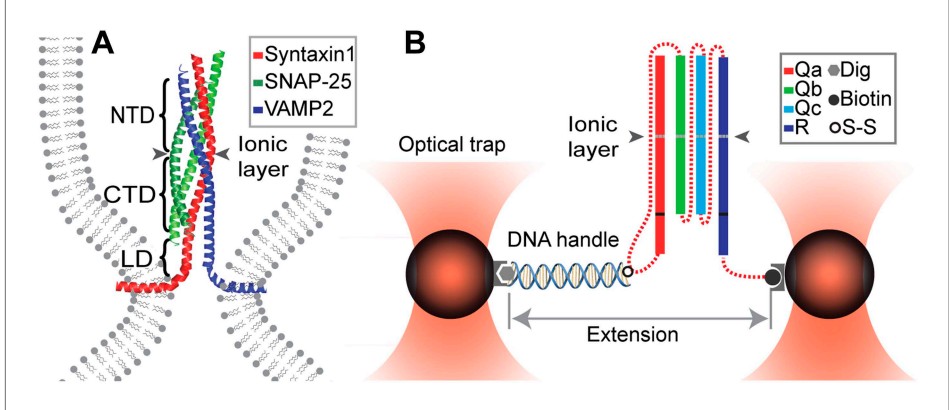

**Figure 1**. Chimeric SNARE construct and experimental setup used to study functional assembly of single SNARE complexes using dual-trap high-resolution optical tweezers. (**A**) Modular parallel four-helix bundle structure of an assembled neuronal SNARE complex mediating membrane fusion. The SNARE complex contains different functional domains: an N-terminal domain (NTD), an ionic layer, a C-terminal domain (CTD), a linker domain (LD), two transmembrane domains, and other domains not shown here. (**B**) Diagram showing the chimeric SNARE construct and the experimental setup. Each SNARE complex contains one SNARE motif from the four highly conserved $Q_a$, $Q_b$, $Q_c$, and R SNARE families. These motifs are joined into one protein through spacer sequences (dashed lines) to facilitate the single-molecule manipulation experiment. The same color coding for different SNARE proteins is used throughout this work. See *Figure 1—figure supplement 1* for complete sequences of the chimeric SNAREs and *Figure 1—figure supplements 2–4* for minimal effects of the spacer sequences on the folding energy and kinetics of the SNARE complexes.

The following figure supplements are available for figure 1:

**Figure supplement 1**. Amino acid sequences of the chimeric SNARE protein constructs used for the single-molecule manipulation study of SNARE assembly.

**Figure supplement 2**. The chimeric neuronal SNARE protein correctly folds into an expected four-helix SNARE bundle.

**Figure supplement 3**. The chimeric neuronal SNARE protein folds into a homogenous SNARE four-helix bundle with an expected molecular weight.

**Figure supplement 4**. The chimeric t-SNARE protein supports lipid mixing between liposomes.

Recently, we have applied high-resolution optical tweezers to quantitatively characterize the energetics and kinetics of neuronal SNARE folding for the first time (*Gao et al., 2012*). This single-molecule manipulation method allows measurement of the folding energy and kinetics of macromolecules under equilibrium conditions (*Liphardt et al., 2001*). Furthermore, the external force applied to the SNARE complex mimics the opposing force from membranes (*Li et al., 2007*; *Liu et al., 2009*; *Min et al., 2013*). Using this single-molecule method, we proved the long-standing hypothesis that neuronal SNAREs assemble by a zippering mechanism and discovered a half-zippered SNARE intermediate that plays a crucial role in the synchronized, calcium-triggered synaptic vesicle fusion (*Gao et al., 2012*). Step-wise SNARE zippering is initiated by slow association between N-terminal domains (NTDs) of t- and v-SNAREs. SNARE assembly then pauses in the half-zippered state in a force-dependent manner. Finally, the C-terminal domain (CTD) of the v-SNARE (VAMP2 or synaptobrevin) rapidly zippers along the pre-structured t-SNARE template to drive fast membrane fusion.

It is unknown whether other SNAREs assemble by the same zippering mechanism. Furthermore, it is not clear how SNARE assembly is adapted to efficient and versatile membrane fusion. It has been proposed that SNAREs generally assemble in an all-or-none manner without any partially folded intermediates (*Jahn and Fasshauer, 2012*; *Kasai et al., 2012*). It is argued that assembly of neuronal SNARE complexes occurs in a large energy gradient, and thus cannot be stopped halfway to form any partially assembled intermediates (*Jahn and Fasshauer, 2012*). However, despite its fast speed,

downhill SNARE assembly would be poorly coupled to membrane fusion, resulting in low energy efficiency of the SNARE engine. In contrast, many molecular engines tested at a single-molecule level have nearly 100% energy efficiency (*Bustamante et al., 2004*). Based on the first law of thermodynamics, a mechanochemical process has 100% energy efficiency only when the process is reversible. Therefore, to maximize their energy efficiency, SNAREs are expected to fold in a relatively smooth energy landscape (*Onuchic and Wolynes, 2004*) in the presence of the membrane load. This requires a close match between the energy landscape of SNARE assembly and the energy profile of membrane interactions. The energy opposing membrane fusion includes contributions from the long-ranged entropic membrane undulation, membrane deformation, and electrostatic interactions, and the short-ranged membrane dehydration and van der Waals interactions (*Leckband and Israelachvili, 2001*). The strong short-ranged repulsion is the largest energy barrier for fusion and takes place within a few nanometers of membrane separation, thus constituting a hard core for fusion. To break this hard core, a SNARE complex is required to focus its folding energy on the membrane proximal C-terminus. Therefore, analogous to a car engine, an efficient SNARE engine is expected to change gears to meet increasing resistance as SNAREs fold towards membranes. However, it remains unclear whether such a gear-changing mechanism exists in SNARE assembly.

To address the above questions, we measured the folding energy and kinetics of four representative SNARE complexes at a single-molecule level, using high-resolution optical tweezers and a new chimeric SNARE design (*Figure 1*). These complexes mediate highly regulated exocytosis of neurotransmitters in pre-synaptic neurons (neuronal SNAREs: syntaxin 1, SNAP-25B, and VAMP2 or synaptobrevin) (*Sollner et al., 1993*) and translocation of glucose transporter type 4 (GLUT4) in adipocytes or muscle cells (GLUT4 SNAREs: syntaxin 4, SNAP-23, and VAMP2) (*Bai et al., 2007*; *Stockli et al., 2011*). The complexes also affect constitutive fusion of endocytic vesicles to early endosome in mammals (endosomal SNAREs: syntaxin 13, Vti1A, syntaxin 6, and VAMP4) (*Zwilling et al., 2007*) and fusion of post-Golgi vesicles to plasma membranes in yeast (yeast SNAREs: Sso1, Sec9, and Snc2) (*Strop et al., 2008*). All four of these SNARE complexes were chosen for our study because they represent SNAREs in diverse evolutionary species, have different degrees of regulation, and mediate fusion with a speed ranging from 0.2 ms to 20 min (*Kasai et al., 2012*). In addition, the crystal structures of neuronal, endosomal, and yeast SNARE complexes are available (*Sutton et al., 1998*; *Zwilling et al., 2007*; *Strop et al., 2008*; *Stein et al., 2009*), which facilitates derivation of their various assembly intermediates from our single-molecule measurements (*Gao et al., 2012*), allowing us to compare the folding pathways and energy landscapes of different SNARE complexes.

Our results show that all four SNARE complexes assemble via the same zippering mechanism in three sequential steps: slow NTD association, fast CTD zippering, and finally rapid LD zippering. However, the CTD zippering energy of different SNARE complexes varies greatly and is highly concentrated at the C-terminus.

## Results

### Chimeric SNARE complex and experimental setup

To facilitate protein preparation and single-molecule experiments, we constructed new chimeric SNARE proteins in which three or four cognate SNARE proteins were joined into one polypeptide with the addition of two or three spacer sequences (*Figure 1*). Individual cytoplasmic SNARE sequences were truncated and regions that directly participate in SNARE complex formation were kept (*Figure 1—figure supplement 1*). To minimize their perturbation on the structure and dynamics of SNARE complexes, the spacer sequences were chosen to be unstructured and of proper length. Each chimeric SNARE protein consisted of a unique cysteine at the C-terminus of $Q_a$ SNARE and an Avi-tag at the C-terminus of R SNARE used to pull the single SNARE complex (*Figure 1B*).

We first examined the structural and functional integrity of the chimeric SNARE complexes. For this purpose, the recombinant proteins were purified and biotinylated in vitro. The expected helical bundles that formed were confirmed by circular dichroism spectra and gel filtration profiles (*Figure 1—figure supplements 2 and 3*). To test the function of the SNARE protein, we similarly made a chimeric neuronal t-SNARE protein and tested its ability to mediate lipid mixing with full-length VAMP2 (*Figure 1—figure supplement 4*). We found that the t-SNARE protein was as fusogenic as the wild-type cytoplasmic t-SNARE complex that is covalently linked to the membrane (*McNew et al., 2000*). This result suggests that the spacer sequence between syntaxin and SNAP-25 does not significantly interfere with SNARE

assembly and membrane fusion. Furthermore, the chimeric neuronal SNARE complex reveals folding energy and kinetics (see below) consistent with our recent reports based on a different SNARE construct in which syntaxin and VAMP2 were cross-linked at their N-termini by a disulfide bond (*Gao et al., 2012*). Taken together, the chimeric SNARE proteins mimic their corresponding SNARE complexes and can be used to facilitate the study of SNARE assembly at a single-molecule level. We refer to these proteins as SNARE complexes.

The SNARE complexes were cross-linked to a 2260 bp DNA handle (*Cecconi et al., 2005*) and tethered to two polystyrene beads held in two optical traps of high-resolution optical tweezers (*Moffitt et al., 2006*; *Sirinakis et al., 2011*; *Figure 1B*). Single SNARE complexes were pulled from the C-termini of $Q_a$ and R SNAREs by moving one trap relative to another at a constant speed, typically 10 nm/s. The tension and extension of the protein-DNA tether were recorded at 10 kHz and used to derive protein folding energy and kinetics.

## Common intermediates and pathways of SNARE assembly

When pulled to a force up to 25 pN, all four SNARE-DNA tethers extended continuously in some force ranges, but discontinuously in other ranges (*Figure 2A,B*). The continuous extension increase was mainly caused by stretching of the semi-flexible DNA handle while the SNARE complex remained in the same folding state. The resultant force-extension curves (FECs) could generally be fit by the worm-like chain model of the DNA and polypeptide (*Marko and Siggia, 1995*). In contrast, abrupt extension changes resulted from cooperative protein transitions between different states (*Figure 2C*). The FECs show that all four SNARE complexes sequentially unfolded via two reversible transitions and one or two irreversible unfolding steps. Compared to the FECs reported for the neuronal SNARE complex (*Gao et al., 2012*) and confirmed by the detailed analysis described below, the second reversible transition (between state 2 and state 3) and the first irreversible transition (between state 3 and state 4) resulted from folding/unfolding transitions of CTD and NTD, respectively. Both transitions are energetically or kinetically distinct for each of the four SNARE complexes, as is demonstrated by non-overlapping distributions of the characteristic forces or different lifetimes associated with these transitions (*Figure 3*). In particular, NTD is mechanically more stable than CTD and unfolded generally after $10–10^5$ CTD folding and unfolding transitions under our experimental conditions (*Figure 2B*).

After the last irreversible unfolding event, the FECs obtained by pulling proteins to higher forces (>25 pN) did not show any additional discontinuous extension changes (*Figure 2A*), indicating that the SNARE complexes had been completely unfolded. When relaxed, the SNARE complex remained unfolded until the force was dropped to ~4 pN, leading to a large hysteresis in the FECs. Further relaxation of the complex to lower forces led to FECs overlapping with those of the FECs in the pulling phase, often with small and sudden extension drops manifesting cooperative reassembly of SNARE complexes (*Figure 4*). Additional cycles of pulling and relaxation generally revealed overlapping FECs, suggesting that the SNARE complexes could fully reassemble into nearly identical structures under our experimental conditions.

The reversible SNARE transitions could be better observed under approximately constant forces (*Figure 5A*, *Figure 6A*). In this case, the time-dependent extension change represented spontaneous folding/unfolding transition of the protein under tension due to thermal fluctuations. Both transitions in each of the four SNARE complexes were binary, as indicated by the two peaks in the histogram distributions of extension (*Figure 5B*, *Figure 6B*). The transitions remained cooperative at all forces tested, but were shifted to unfolded states at higher forces. Furthermore, the four SNARE complexes had similar average extension changes for both transitions (*Table 1*), implying that the same SNARE domains were involved in the observed transitions. Taken together, the results from experiments in variable and constant forces suggest that all four SNARE complexes follow similar pathways to assembly or disassemble via at least two intermediates.

To derive the structures of the intermediates observed in our experiments, we fit the continuous regions of the FECs using a quantitative model of the protein-DNA conjugate previously reported (*Gao et al., 2012*; *Xi et al., 2012*). In this model, the extension of the structured portion of the SNARE complex is force-independent, but varies as the SNARE complex changes its conformation ('Materials and methods'). The model generally fit the measured FECs and extension changes obtained at constant forces well (*Figure 2A*). Extensive analysis revealed two common intermediates for the four SNARE complexes: the LD-unfolded state and the half-zippered state (*Figure 2C*). In the former, the SNARE LD was unfolded, while its CTD remained approximately intact. In the latter, the C-terminal half

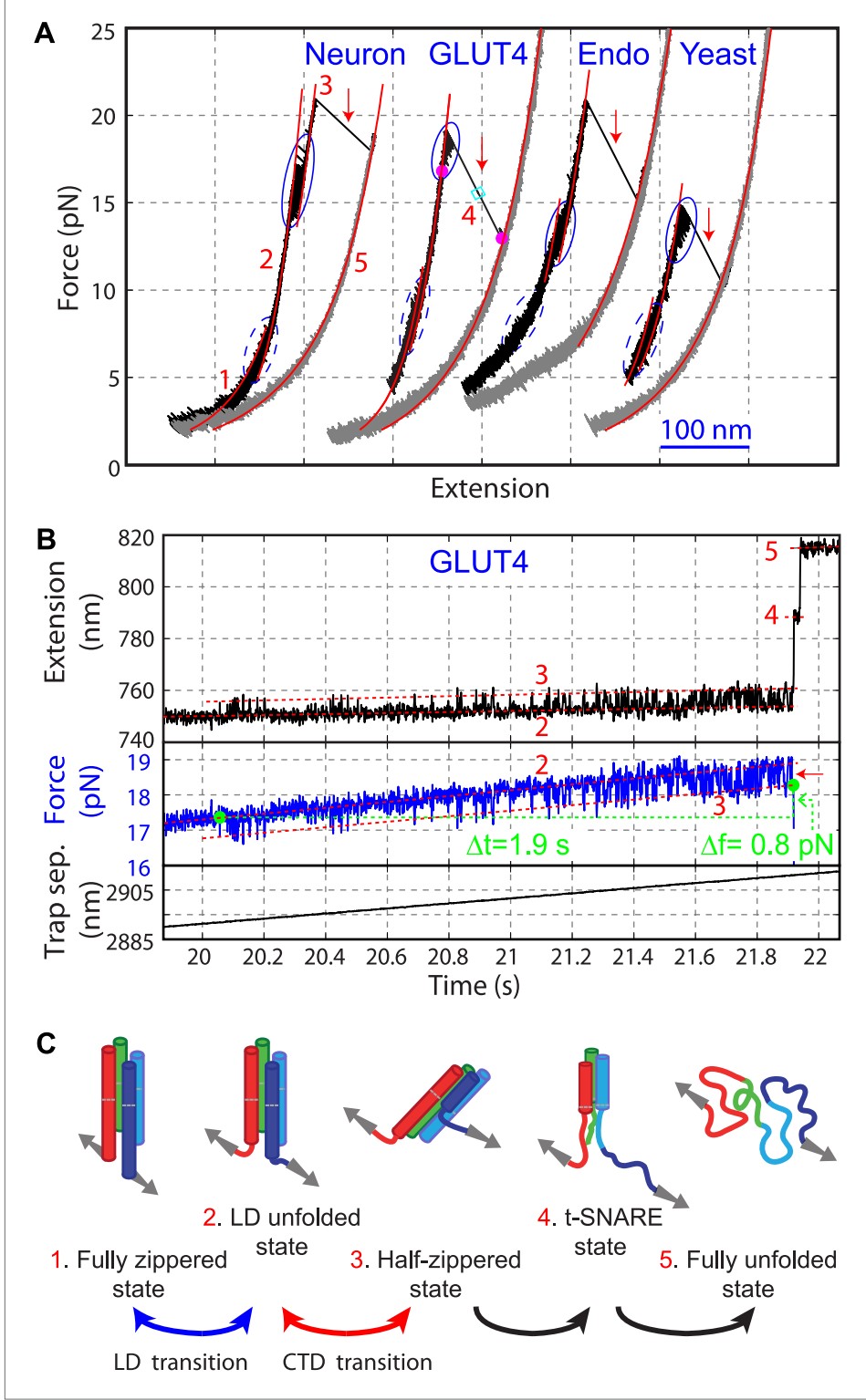

**Figure 2**. Four representative SNARE complexes assemble or disassemble via common intermediates and pathways. (**A**) Force-extension curves (FECs) of the neuronal, GLUT4, endosomal, and yeast SNARE complexes. FECs were obtained by pulling the complexes (black) or relaxing them (gray). The reversible C-terminal domain (CTD) and linker domain (LD) folding/unfolding transitions are marked by blue solid and dashed ovals, respectively, whereas irreversible unfolding of the partially zippered SNARE complex is indicated by a red arrow. Continuous

*Figure 2. Continued on next page*

*Figure 2. Continued*

FEC regions can be fit by the worm-like chain model and represent different SNARE states numbered as in (**C**). Below ~6 pN, deviation of some fits from the measured FECs corresponding to the unfolded complex (state 5) may be caused by intramolecular interactions or partial refolding of the complex. The t-SNARE state can be identified from some FECs, with a transient one (~20 ms) marked by a cyan rectangle. (**B**) Time-dependent extension, force, and trap separation corresponding to the CTD and N-terminal domain (NTD) transition region in the FEC of the GLUT4 SNARE complex in A (marked by two magenta dots). In the upper and middle panels, the positions of different SNARE folding states are indicated by red dashed lines. About 90 CTD transitions occurred before NTD unzipping and reaching ~18.6 pN equilibrium force (indicated by a red arrow in the middle panel). Here the equilibrium force for a two-state protein folding/unfolding process is defined as the average state forces (marked by dashed lines) under which the folded and the unfolded states are equally populated. Note that most NTD unzipping took place in the CTD-unfolded state (state 3). In the middle panel, the first CTD and the first NTD unzipping events during the pulling process are indicated by green dots and their time and force differences indicated. The time and force distributions are shown in *Figure 3B,C*. In the bottom panel, the separation between two optical traps was increasing at a speed of 10 nm/s to slowly pull the single SNARE complex. (**C**) Different SNARE assembly states partly derived from model-fitting of FECs shown in (**A**). Gray arrows indicate the pulling direction. Data associated with all FECs shown in this work were mean-filtered using 5 ms time window. See more FECs and their associated features in *Figure 2—figure supplements 1–4*.

The following figure supplements are available for figure 2:

**Figure supplement 1**. Distinct linker domain and C-terminal domain transitions.

**Figure supplement 2**. The neuronal t-SNARE complex as a transient unfolding intermediate of the half-zippered SNARE complex.

**Figure supplement 3**. The GLUT4 t-SNARE complex as a transient unfolding intermediate of the half-zippered SNARE complex.

**Figure supplement 4**. The yeast t-SNARE complex is a stable unfolding intermediate of the half-zippered SNARE complex.

of the R SNARE was unzipped, whereas three Q SNARE motifs remained intact (*Kummel et al., 2011*; *Gao et al., 2012*; *Li et al., 2014*). Specifically, neuronal, GLUT4, endosomal, and yeast SNARE complexes in the half-zippered state had their R SNAREs unzipped to −1, +3, +1, and +3 amino acids relative to the ionic layer, respectively, where the positive sign designates the C-terminal amino acids. The standard deviation of all positions was less than three amino acids (*Table 1*).

To further confirm the derived structures of the intermediate states, we truncated the LD or the CTD of the v-SNARE Snc2 in the yeast SNARE complex and repeated the pulling experiment. We found that LD truncation eliminated the LD but not the CTD transition, while the CTD truncation abolished both transitions (*Figure 2—figure supplement 1*). These results support the inferred structures for the intermediate states. Finally, both CTD and LD folded more rapidly than similar coiled-coil proteins (*Xi et al., 2012*), with their transition rates greater than 50 s$^{-1}$, even at the equilibrium forces (*Figure 5*, *Figure 6*).

Further unzipping of the half-zippered states of all four SNARE complexes became irreversible and they remained unfolded for over 50 s under the slow relaxation conditions in our experiment (*Figure 2A*), indicating a large energy barrier for SNARE NTD association. Close inspection of the FECs showed that a fraction of half-zippered SNARE complexes, that is, 10%, 50%, and 30% for neuronal, GLUT4, and endosomal SNAREs, respectively, unfolded via a transient intermediate with a typical lifetime of less than 50 ms (*Figure 2A,B*, *Figure 2—figure supplements 2 and 3*). The yeast SNARE complex is special, because this additional intermediate appeared in 85% of the unfolding transitions of the half-zippered complex and generally lasted for more than 5 s (*Figure 2—figure supplement 4*). For all SNARE complexes, these intermediate states are located at an extension approximately halfway between the half-zippered states and the fully unfolded states, indicating their similar structures. Based on their relative extension positions, the intermediate states are estimated to be t-SNARE or Q SNARE complexes with ordered NTDs but disordered CTDs (*Figure 2C*).

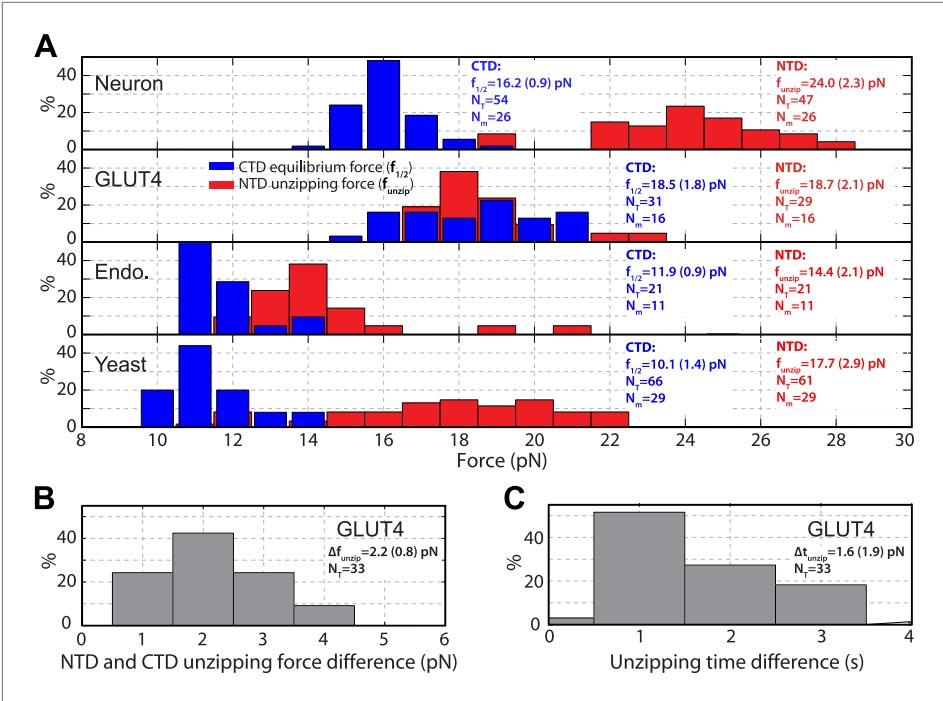

**Figure 3**. Distinct transition kinetics and stabilities of SNARE C-terminal domain and N-terminal domain. (**A**) Histogram distributions of the C-terminal domain (CTD) equilibrium force and the N-terminal domain (NTD) unzipping force for different SNARE complexes. The vertical axis shows the percentage of the event number in each bin. The average CTD equilibrium force ($f_{1/2}$) or NTD unzipping force ($f_{unzip}$) scored on the total numbers of transition events ($N_T$) and single SNARE complexes ($N_m$) are indicated, with the number in parenthesis designating the standard deviation of the mean. (**B, C**) Histogram distributions of the force and time differences of the first NTD and CTD unzipping events (*Figure 2B*). The average force difference ($\Delta f_{unzip}$) or time difference ($\Delta t_{unzip}$) is indicated. The distinct CTD and NTD transition kinetics are revealed by non-overlapping force distributions for neuronal, endosomal, and yeast SNARE complexes or significant force and time differences associated with the first unzipping events of CTD and NTD of the GLUT4 SNARE complex. Note that optical tweezers have a force measurement accuracy of 10% absolute forces between different single molecules and of <0.1 pN relative forces within same single molecules (*Moffitt et al., 2006*).

In conclusion, all four SNARE complexes assemble or disassemble via three common intermediate states: the t-SNARE state, the half-zippered state, and the LD-unfolded state, and along similar folding pathways and kinetics, particularly slow NTD association and fast CTD and LD zippering.

## Energetics and kinetics of CTD zippering

The biggest difference between the four SNARE complexes lay in their CTD equilibrium forces (*Figure 3A*, *Figure 5C*), indicating different CTD folding energies. To quantify CTD folding energy and kinetics, we measured CTD transitions at different constant forces in their corresponding force ranges (*Figure 5A*). We analyzed each extension-time series using a two-state hidden Markov model (HMM) and determined the positions of the folded and unfolded CTD states and their corresponding fluctuations, the unfolding probability, and transition rates (*Gao et al., 2012*; *Xi et al., 2012*). The HMM-based analyses yielded idealized state transitions and extension histogram distributions that closely matched the corresponding experimental measurements (*Figure 5A,B*). The unfolding probability rises with the force increase in a sigmoidal manner (*Figure 5C*). The folding rate or unfolding rate decreases or increases approximately exponentially upon a force increase in the narrow force range tested (*Bustamante et al., 2004*; *Figure 5D*). Both observations suggest a two-state CTD transition and the existence of a single major energy barrier corresponding to the transition state for the folding/unfolding process. The position of the transition state relative to the folded or unfolded state can be determined from the force-dependent transition rates.

We adopted a simplified energy landscape model to derive the energy and rate of SNARE folding at zero force ('Materials and methods') (*Gao et al., 2012*; *Xi et al., 2012*). Non-linear least-squares fitting

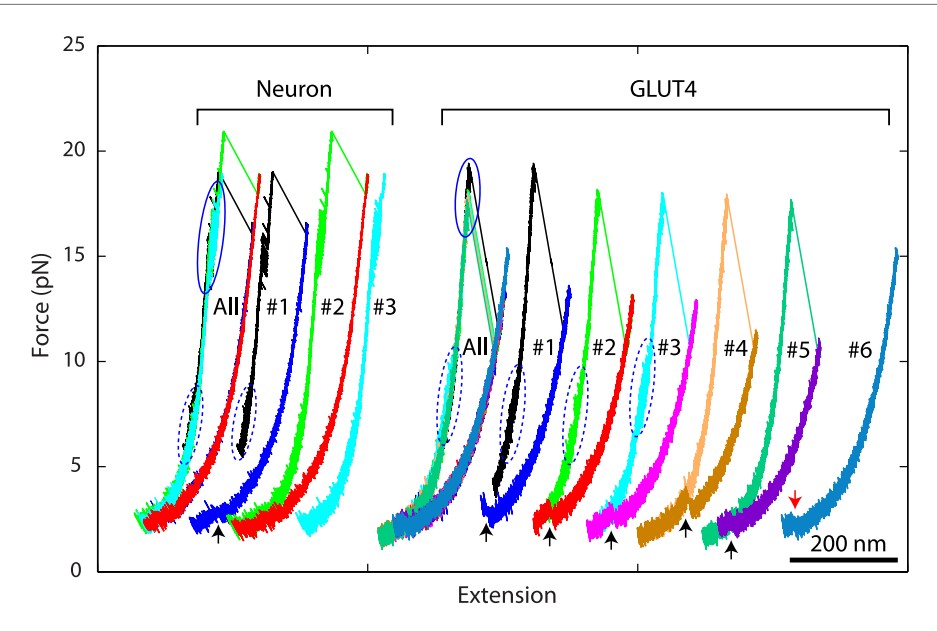

**Figure 4**. Overlapping force-extension curves obtained by repeatedly pulling a single neuronal or GLUT4 SNARE complex, revealing robust and common step-wise SNARE assembly and disassembly. The overlapping force-extension curves (FECs) (designated by 'All') are shifted along the x-axis to reveal individual FECs corresponding to different pulling cycles (numbered). The cooperative reassembly events are indicated by black arrows. The neuronal SNARE-DNA tether broke in the third pulling cycle of the neuronal SNARE complex at the maximum pulling force. The GLUT4 SNARE complex unfolded at 2.5 pN force (red arrow) in the last pulling cycle, indicating that the complex was not properly assembled at the end of the fifth pulling cycle. Note that heterogeneity in SNARE zippering was observed, a phenomenon also seen in many single-molecule experiments (*Lu et al., 1998*; *Sirinakis et al., 2011*). The heterogeneity is manifested by changes in the rate and/or the equilibrium force of the C-terminal domain (CTD) transition detected in different pulling cycles of the same chimeric SNARE protein. For the single neuronal SNARE protein shown here, both equilibrium force and rate of the CTD transition are slightly lower in the first pulling cycle than in the following two cycles. More heterogeneity can be seen in *Figure 2—figure supplements 2 and 3*.

of the model matched the experimental data well (*Figure 5C,D*), which revealed the free energy of the folded state and the transition state and their relative positions (*Table 1*). The CTD folding energy of neuronal, GLUT4, endosomal, and yeast SNARE complexes were −27 (±5; SD throughout the text) $k_BT$, −23 (±4) $k_BT$, −16 (±2) $k_BT$, and −13 (±3) $k_BT$, respectively. The CTD folding energy and the equilibrium rate (~100 s$^{-1}$) of the neuronal SNARE complex were very close to the energy (28 ± 3 $k_BT$) and the rate (~160 s$^{-1}$) reported earlier (*Gao et al., 2012*), indicating that the spacer sequences in the chimeric construct used here have minimal effect on the folding energy and kinetics of the SNARE complex.

The binary CTD transition manifested the existence of an energy barrier and its associated transition state for CTD folding and unfolding in the presence of the external force. When extrapolated to zero force, the CTD folding energy barrier became minimal for endosomal and yeast SNARE complexes or disappears for neuronal and GLUT4 SNARE complexes (*Table 1*). In both scenarios, free energy of the transition states can still be defined (*Gao et al., 2012*; *Xi et al., 2012*). The transition states of four SNARE complexes are located between the third and sixth hydrophobic layers. The energy and position of the transition state is important for characterizing the energy landscape of SNARE folding described later in the text. The relatively small folding energy barrier suggests that the rate of SNARE-mediated fusion is not limited by the intrinsic rate of CTD folding (at zero force), and that the stability of the half-zippered state is strongly force-dependent. Any partially zippered trans-SNARE complexes involved in vesicle docking and priming are likely in strained states imposed by the membranes and regulatory proteins (*Guzman et al., 2010*; *Jahn and Fasshauer, 2012*).

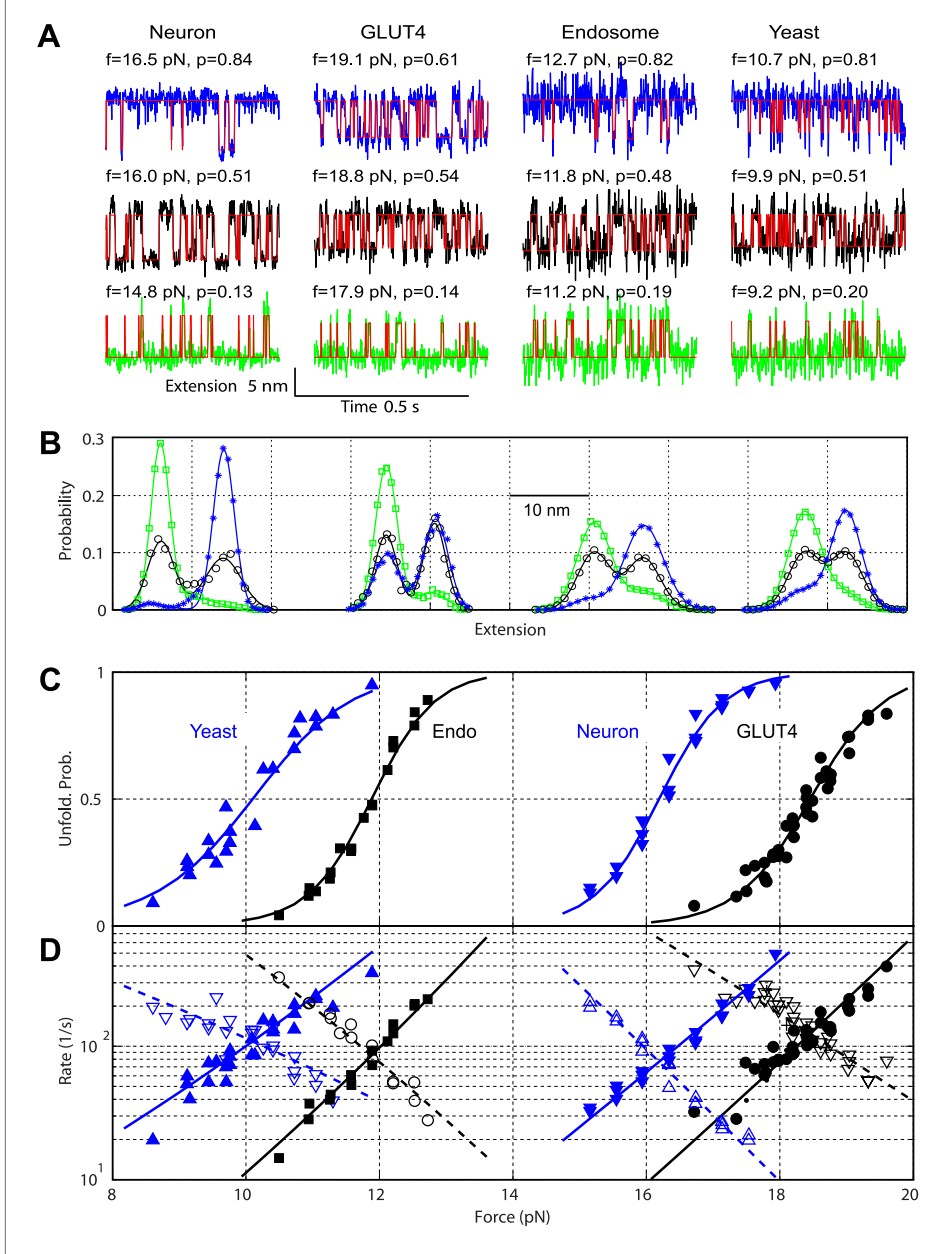

**Figure 5**. Comparison of the two-state C-terminal domain transitions of four SNARE complexes. (**A**) Force-dependent extension-time trajectories under approximately constant forces (f) revealing the unfolding probability (p) of C-terminal domain (CTD) as indicated. The idealized two-state transitions (red lines) were calculated based on a hidden Markov model (HMM). (**B**) Histogram distributions of the extensions shown in **A** (symbols) and their best fits with double-Gaussian functions (lines). For best comparison, the distributions for each SNARE complex were shifted along the x-axis to align them at the same average position of the unfolded CTD state. Distributions at different forces are color-coded as the corresponding extension traces in **A**. All the extension-time trajectories shown in this work were mean-filtered using a 1 ms time window. (**C**) CTD unfolding probabilities of four SNARE complexes. (**D**) The corresponding folding rates (hollow symbols) and unfolding rates (solid symbols) of CTD transitions. The best-fit unfolding probability (solid line), folding rate (dashed line), and unfolding rate (solid line) were obtained by non-linear least-squares fitting using a simplified energy-landscape model of SNARE assembly ('Materials and methods').

## Energetics and kinetics of LD zippering

The folding and unfolding transitions of the LD in four SNARE complexes were similar. They were reversible, binary, and fast (*Figure 6*, *Figure 6—figure supplement 1*). Furthermore, the LD transitions

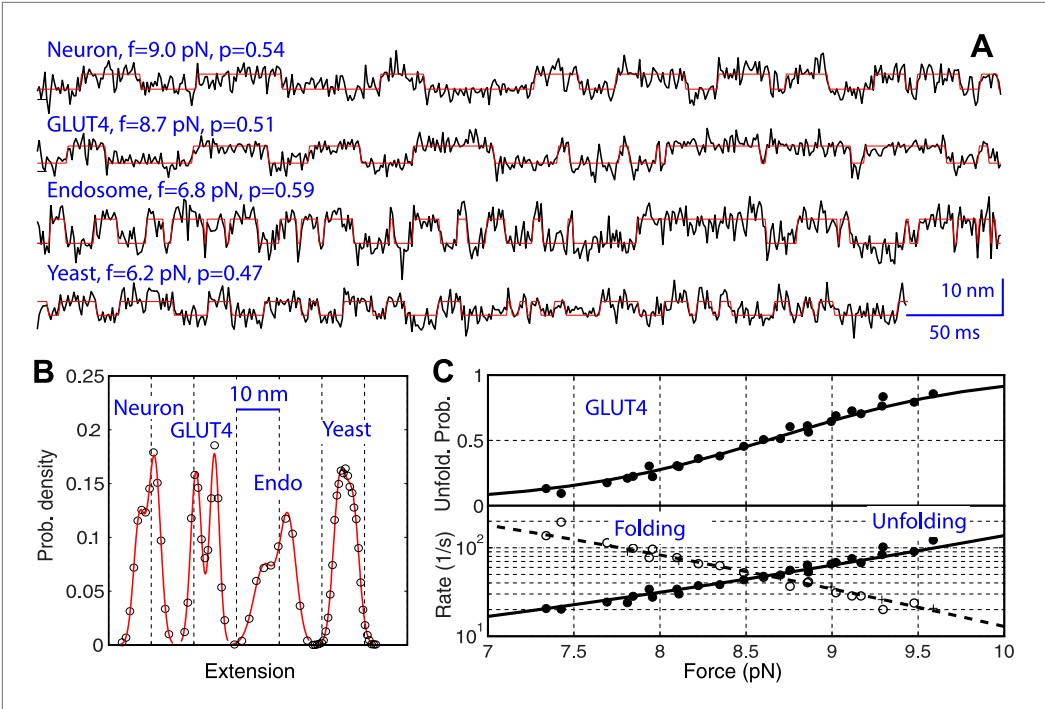

**Figure 6**. Comparison of the two-state linker domain transitions of four SNARE complexes. (**A**) Extension-time trajectories (black lines) and their best hidden Markov model (HMM) fits (red lines) showing fast binary transitions of linker domains (LDs) under constant forces. The force (f) and unfolding probability (p) are indicated. (**B**) Extension histogram distributions corresponding to the trajectories in **A** (symbols) and their best fits with double-Gaussian functions (red lines). (**C**) Force-dependent unfolding probability and transition rates (symbols) and their best fits (solid or dashed lines) of GLUT LD. Similar data corresponding to other SNARE complexes are shown in *Figure 6—figure supplement 1*.

The following figure supplements are available for figure 6:

**Figure supplement 1**. Folding energy and kinetics of SNARE linker domains.

**Figure supplement 2**. Minor effect of the spacer sequences in the chimeric SNARE proteins on the folding energy of SNARE complexes.

occurred in narrow force ranges (6–8.6 pN), in contrast to the CTD transition (10.1–18.5 pN) (*Table 1*). Although the LD of the endosomal SNARE complex forms a four-helix bundle (*Zwilling et al., 2007*; *Figure 1—figure supplement 1*), rather than a two-stranded coiled coil as in the other three SNARE complexes, it has similar LD transition kinetics, associated extension change, and lower equilibrium force than its CTD. This observation corroborates the conclusion that LD is a domain distinct from CTD, even in the endosomal SNARE complex.

The energy and kinetics of LD zippering at zero force was obtained in a way similar to CTD, as previously described (*Gao et al., 2012*). For neuronal SNARE complexes, the new chimeric construct led to an equilibrium force of 8 (±1) pN for LD transition, compared to 12 (±2) pN previously measured for the same transition. Correcting for the minor effect of the spacer sequence added between syntaxin and SNAP-25 (*Figure 6—figure supplement 2*), we obtained LD zippering energy of −10 (±2) $k_BT$ for the neuronal SNARE complex, consistent with our previous measurement of −8 (±2) $k_BT$. Similarly, we derived the zippering energy of LDs and their associated energy barriers for the other three SNARE complexes (*Table 1*). In the four SNARE complexes, LD zippering outputs less energy than CTD zippering. Thus, CTD zippering serves as the major power stoke for membrane fusion (*Walter et al., 2010*).

## Different roles of SNARE zippering stages in membrane fusion

Our above analysis revealed a simplified folding energy landscape of each SNARE complex (*Figure 7*). To illustrate how such an energy landscape is adapted to stage-wise membrane fusion (*Figure 7A*), we

**Table 1.** Average equilibrium force, extension change, folding energy, and folding energy barrier and position associated with C-terminal domain and linker domain transitions of the four different SNARE complexes

| | C-terminal domain | | | | | Linker domain | | | | |
|---|---|---|---|---|---|---|---|---|---|---|
| SNARE complex | Force (pN) | Extension change (nm) | Folding energy ($k_BT$) | Transition state energy* ($k_BT$) | Transition state position† (a.a.) | Force (pN) | Extension change (nm) | Folding energy ($k_BT$) | Transition state energy* ($k_BT$) | Transition state position† (a.a.) |
| Neuron | 16.2 (0.9) | 7.2 (1.2) | −27 (4.7) | −5.5 (1.5) | 17 (3) | 8 (1) | 4.7 (0.5) | −9.7 (1.6) | 5.5 (1.5) | 31 (1) |
| GLUT4 | 18.5 (1.8) | 6.0 (0.9) | −23 (4.1) | −0.8 (1.0) | 11 (2) | 8.6 (0.9) | 5.6 (1.1) | −12 (2.7) | 2 (1.0) | 30 (1) |
| Endosome | 11.9 (0.9) | 6.9 (0.4) | −16 (1.5) | 2.1 (1.4) | 12 (2) | 6.3 (1.2) | 5.1 (1.8) | −6.1 (2.4) | 4.9 (1.5) | 32 (2) |
| Yeast | 10.1 (1.4) | 5.8 (0.8) | −13 (2.5) | 3.2 (1.5) | 13 (2) | 6.0 (1.6) | 5.1 (1.2) | −5.7 (2.0) | 3.6 (2.0) | 32 (2) |

*Here, negative energy indicates downhill protein folding (**Yang and Gruebele, 2003**).
†The number of the amino acids in the R SNARE C-terminal to the ionic layer (chosen as 0).
The equilibrium force and extension change were determined at an unfolding probability of 0.5 for the two-state processes. The standard deviations of the averages are shown in parenthesis. The equilibrium force distribution, the number of transitions, and the number of single molecules scored for C-terminal domain (CTD) transitions are shown in **Figure 3**. For parameters related to linker domain (LD) transitions, a total of 18, 35, 11, and 24 LD transitions in single neuronal, GLUT4, endosomal, and yeast SNARE complexes were scored, respectively.

calculated the energy landscape of SNARE assembly in the presence of membranes using a neuronal SNARE complex as an example (**Gao et al., 2012**). The interaction energy between membranes containing lipid-anchored t- and v-SNAREs has been measured by the surface forces apparatus (SFA) (**Li et al., 2007**). The interaction as a function of membrane separation contains two exponentially decaying components with decay constants of 2.5 nm ($d_1$) and 6 nm ($d_2$). The short-ranged component represents membrane repulsion just before fusion, including membrane dehydration (**Leckband and Israelachvili, 2001**), and the long-range component results from the steric repulsion between unfolded or partially unfolded t- and v-SNAREs before their association.

We chose the membrane interaction energy ($V$) per SNARE complex versus the distance between two membrane surfaces at the sites of SNARE attachment ($d$) as

$$V(d) = \frac{E_m}{1+\alpha}\left[\exp\left(-\frac{d-d_c}{d_1}\right) + \alpha \exp\left(-\frac{d-d_c}{d_2}\right)\right], d \geq d_c, \tag{1}$$

where $E_m$ determines the energy barrier for membrane fusion per SNARE complex when membranes are brought to the minimal distance allowed by the molecular dimension of the fully folded SNARE four-helix bundle ($d_c$ = 1 nm) (**Sutton et al., 1998**; **Stein et al., 2009**; **Figure 7B**). Below this critical distance, membrane fusion occurs irreversibly. The amplitude ratio of the two exponential components ($\alpha$) was set to 0.5 based on the SFA measurement (**Li et al., 2007**). The energy barrier for membrane fusion ($N \times E_m$) and the exact number of SNARE complexes required for fusion ($N$) are under much discussion (**Karatekin et al., 2010**; **Mohrmann et al., 2010**; **van den Bogaart et al., 2010**). To bypass these uncertainties, we chose $E_m \approx 50\ k_BT$ per SNARE complex, consistent with our measured folding energy per neuronal SNARE complex and other estimations (**Li et al., 2007**; **van den Bogaart et al., 2010**).

The role of the LD in membrane fusion is not clear. Evidence suggests that LDs bind membranes and constitute parts of membrane anchors with SNARE transmembrane domains (**Li et al., 2007**; **Ellena et al., 2009**; **Borisovska et al., 2012**). To simplify our calculations, we did not explicitly consider extension and energy contributions from LDs but assumed instead that membrane fusion occurs when CTD is fully zippered.

We computed the energy landscape of the loaded SNARE complex as the sum of the energy of the unloaded SNAREs, the entropic energy of the stretched and unfolded SNARE polypeptides calculated based on **Equation 3** in 'Materials and methods', and the membrane interaction energy. At each SNARE zippering stage, an equilibrium membrane distance was calculated by equating the SNARE pulling force to the membrane repulsive force (**Figure 7C,D**). The calculated SNARE energy was plotted in **Figure 7B** as a function of the membrane distance. SNARE NTD association was initiated

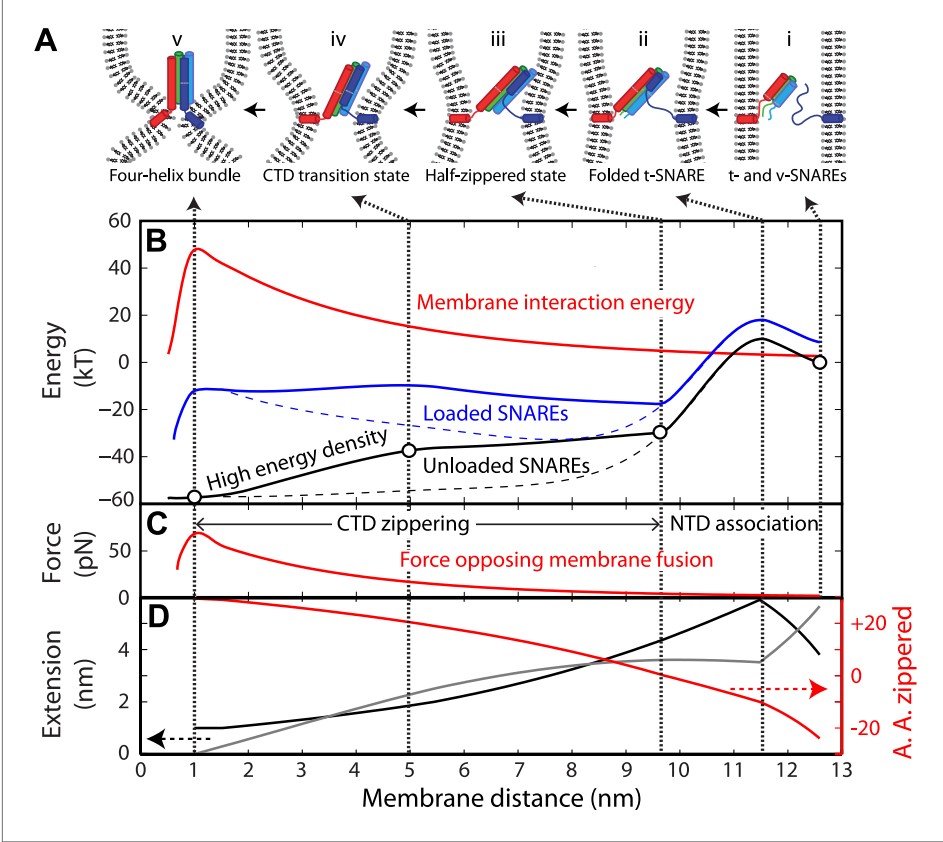

**Figure 7**. Energy landscape of SNARE zippering perfectly meets the needs of membrane fusion. (**A**) Cartoons of different assembly states of the trans-SNARE complex corresponding to the points indicated in **B**. (**B**) Free energy of membrane fusion per SNARE complex (red line), a single loaded trans-SNARE complex (blue), or a single unloaded SNARE complex (black) as a function of the distance between two membrane surfaces. The experimental and alternative energy landscapes are plotted in solid and dashed lines, respectively. For the unloaded SNARE complex, the free energy at each membrane distance represents the energy of the SNARE complex in the same zippering state as the trans-SNARE complex at that distance. The experimental energy landscape of the unloaded SNARE complex was derived from the measured energy at characteristic points (marked by circles) through interpolation. (**C**) Repulsive force between two membranes opposing their fusion. (**D**) Extension of all the unfolded amino acids in $Q_a$ and R SNAREs under membrane tension (gray in the left axis) or the folded portion of the SNARE complex (black) and zippering stage of the amino acids (A.A.) in R SNARE (red line in the right red axis). The amino acid number indicates the position of the amino acid relative to the ionic layer (0). The amino acids with negative numbers are in N-terminal domain (NTD) and those with positive numbers in C-terminal domain (CTD). At each amino acid number in the right axis, the R SNARE motif has assembled from its N-terminus (A.A. at −24) to the amino acid with this number. Note that NTD association is accompanied by a relatively small change in membrane distance compared to CTD zippering, because the extension decrease due to NTD folding is largely canceled by the extension increase of the folded t-SNARE.

The following figure supplement is available for figure 7:

**Figure supplement 1**. Estimation of the average forces generated by zippering of the N-terminal and C-terminal CTD of neuronal SNARE complex.

at the very N-termini of t- and v-SNAREs at a large distance of 12.5 nm (**Figure 7B–D**). The association was accompanied by coil-to-helix propagation of the partially disordered t-SNARE towards its C-terminus (**Li et al., 2014**; **Figure 7A**, state ii). Further NTD zippering led to the half-zippered trans-SNARE complex at 9.7 nm (state iii) (**Bharat et al., 2014**). Thus, NTD association occurs in a narrow distance range of 9.7–12.5 nm, where the membrane repulsive force is small (2–5 pN). Formation of the half-zippered state in the presence of membranes leads to a net energy release of ~26 $k_BT$ (the energy difference between state i and state iii) (**Gao et al., 2012**), which can be used to dock or prime vesicles and prevent dissociation of the half-zippered trans-SNARE complex.

In contrast to NTD association, CTD zippering from the half-zippered state was directly and tightly coupled to membrane fusion (*Figure 7B–D*). The membrane-loaded half-zippered SNARE complex had an energy of ~34 $k_BT$ relative to its folded state, which consists of ~27 $k_BT$ CTD folding energy and ~7 $k_BT$ entropic energy stored in the stretched VAMP2 CTD. CTD zippering drew two membranes from 9.7 nm to 1 nm for fusion against a large average force. As a result, CTD zippering of the trans-SNARE complex reduced the energy barrier of membrane fusion (*Figure 7A*, state iv) to 7.4 $k_BT$ per SNARE complex (the energy difference between state iv and state iii), consistent with a fusion rate of ~600 s⁻¹, where we assumed a maximum fusion rate of $10^6$ s⁻¹ in the absence of any energy barrier (*Yang and Gruebele, 2003*; *Gao et al., 2012*). Correspondingly, the metastable half-zippered SNARE complex in the absence of a force load was stabilized by the short-ranged membrane opposing force (*Figure 7B*) together with regulatory proteins, such as complexin (*Sudhof and Rothman, 2009*; *Gao et al., 2012*; *Min et al., 2013*). In contrast, the optical trapping force used in our experiments was long-range, with a typical force constant of 0.1 pN/nm, compared to an average force constant of 7.1 pN/nm for the membrane force opposing CTD assembly. As a result, the half-zippered SNARE state was typically short-lived (<0.2 ms) upon reassembly of the SNARE complex at the low forces favoring NTD association (*Figure 4*), because the trapping force opposing CTD zippering remained small immediately after NTD was zippered. Thus, the membrane's repulsive force was an integral component of SNARE assembly and regulation.

The exponential increase in the membrane repulsive force below 5 nm (from 17 pN to 60 pN, *Figure 7C*) required an increasing force output as CTD zippers toward its C-terminus. SNARE zippering indeed met this requirement by producing a high force in this region. Here, the magnitude of local force generated by SNARE zippering was equivalent to the slope of the energy landscape of the unloaded SNARE complex with respect to extension, which also represents the energy density (defined as the folding energy per unit length of R SNARE polypeptide chain zippered) distributed along the SNARE bundle. While zippering of the first two-thirds of CTD generated an average force of 8 pN, zippering of the last one-third of CTD produces an average force of up to 32 pN (*Figure 7—figure supplement 1*), well suited to countering the short-ranged membrane opposing force. The position where CTD changed its energy density (*Figure 7B*, at 5 nm) was intriguing, because it was close to the energy barrier of the trans-SNARE complex. This position overlap is no accident, because any force applied to the SNARE complex tilts the CTD zippering energy landscape of the unloaded SNARE complex toward the unfolded state (*Bustamante et al., 2004*), which tends to make the density-changing point an energy barrier (for example, see Figure S9 in *Xi et al., 2012*). It is this energy barrier that results in the binary CTD transition and the polarized CTD energy distribution.

To further corroborate the essential role of the polarized CTD energy distribution in membrane fusion, we calculated the energy landscape of the trans-SNARE complex based on an alternative energy landscape of SNARE zippering (*Figure 7B*, black dashed line). In this alternative landscape, the energy of CTD is enriched at its N-terminus, rather than its C-terminus, but with the same total CTD zippering energy. The half-zippered trans-SNARE complex is now greatly stabilized by membranes (at 8 nm of the blue dashed line), but nearly unable to fuse them, because the energy barrier for fusion increases to 24 $k_BT$ (the energy difference between states at membrane distances of 1 nm and 8 nm) compared to 7.4 $k_BT$ for the wild-type SNARE complex. In this case, significant energy from CTD zippering is not transmitted to membranes, but dissipated as heat. In addition, no additional energy barrier appears before fusion. Similarly, SNAREs alone with this alternative folding energy landscape are expected to zipper or unzip in a continuous manner in response to the force exerted by optical tweezers, in contrast to the observed cooperative two-state manner.

Various parameters for the membrane interaction energy have been reported for different model membranes (*Leckband and Israelachvili, 2001*). To test how variation in membrane properties may change the requirement for the polarized energy distribution, we repeated our above calculations by changing the parameters in *Equation 1* with $L_1$ and $\alpha$ ranging from 1 nm to 3 nm and from 0 to 0.5, respectively. Although energy landscapes of the loaded SNARE complex quantitatively change with these parameters, the energy landscape with a C-terminal polarized energy distribution always led to a much lower energy barrier for fusion than the alternative energy landscapes with an N-terminal polarized energy distribution. Thus, a C-terminal polarized energy distribution is a general requirement for efficient membrane fusion.

Taken together, our observed binary CTD transition indicates that the polarized CTD energy distribution is essential for efficient membrane fusion. In contrast, continuous and progressive SNARE assembly leads to poor coupling to membrane fusion.

## Discussion

### Role of the half-zippered SNARE state in membrane fusion

The identification of half-zippered intermediates in all four representative SNARE complexes suggests that SNARE complexes follow a common zippering mechanism to drive membrane fusion (*Hanson et al., 1997*). This observation is not consistent with alternative mechanisms by which the entire SNARE four-helix bundle assembles in an all-or-none manner (*Jahn and Fasshauer, 2012*; *Kasai et al., 2012*) or in a continuous layer-by-layer manner. Instead, our data reveal two distinct cooperative assemblies of the NTD and CTD in the SNARE complex, which clarifies the detailed zippering kinetics.

The presence of a partially zippered SNARE complex has been supported by many experiments (*Xu et al., 1999*; *Schwartz and Merz, 2009*; *Walter et al., 2010*; *Diao et al., 2012*; *Gao et al., 2012*; *Min et al., 2013*). Our work further demonstrates that the partially zippered complex is a half-zippered complex intrinsic to a SNARE complex and functionally important for membrane fusion. Mutations and truncations that alter the structure of the half-zippered neuronal SNARE complex and/or its folding energy and kinetics abolish membrane fusion (Ma L, Gao Y, Yang G, and Zhang YL, manuscript in preparation). Thus, such a half-zippered SNARE complex is required for fast and regulated synaptic vesicle fusion (*Kummel et al., 2011*; *Li et al., 2011*, *2014*).

Our finding suggests that the half-zippered structure is more ancient in SNARE evolution than any regulators that target this structure, and may have more conserved function in membrane fusion than previously thought. We propose that the step-wise assembly enables reversible folding of SNARE complexes, as shown in our calculations. The step-wise and reversible assembly enhances not only the coupling between SNARE zippering and membrane fusion, but also the specificity of SNARE pairing. Furthermore, the half-zippered SNARE complexes may be the target of their cognate Sec1p/Munc18 (SM)-family proteins essential for SNARE-mediated membrane fusion (*Shen et al., 2007*; *Sudhof and Rothman, 2009*; *Jorgacevski et al., 2011*; *Ma et al., 2013*).

### Role of CTD zippering energy in the rate and mechanism of membrane fusion

For the neuronal SNARE complex, we and others suggested that assembly of the NTD and the CTD has distinct functions: while NTD assembly is responsible for vesicle docking and priming, CTD zippering directly drives membrane fusion (*Walter et al., 2010*; *Gao et al., 2012*). Fusion of GLUT4-storage vesicles (GSVs) with the plasma membrane mediated by the GLUT4 SNARE complex appears to be very similar to fusion of synaptic vesicles, including distinct stages of vesicle docking, priming, fusion (*Stockli et al., 2011*), and close CTD zippering energy. However, the docked GSVs take about 1 min to fuse after insulin triggering (*Bai et al., 2007*). In this case, the observed fusion rate is probably limited by the slow NTD association, but not the fast CTD zippering. Thus, insulin may mainly regulate steps upstream of NTD association.

The CTD zippering energy puts a strong constraint on the detailed mechanism of membrane fusion, including on the number of SNARE complexes required for fusion ($N$). If the total CTD zippering energy of $N$ trans-SNARE complexes is used to lower the energy barrier of membrane fusion ($E_b$) (*Montecucco et al., 2005*; *Mohrmann et al., 2010*), the fusion rate ($k$) should be an exponential function of the total zippering energy, that is, $k = k_0 \times \exp(N \times E_{CTD} - E_b)$, where $E_{CTD}$ is the CTD zippering energy per SNARE complex and $k_0$ a pre-constant. The large difference of CTD zippering energy between either endosomal or yeast SNARE complex and neuronal SNARE complex (12 or 14 $k_BT$) suggests that more endosomal or yeast SNARE complexes than neuronal SNARE complexes may be required to mediate fusion (*Mohrmann et al., 2010*; *Wickner, 2010*; *Shi et al., 2012*).

SNARE-mediated membrane fusion in vivo involves fixed numbers of SNARE complexes characteristic of different fusion processes, likely controlled by regulatory proteins (*Montecucco et al., 2005*). This result is in contrast with reconstituted SNARE-mediated membrane fusion in vitro, in which the SNARE number is probably not controlled (*Karatekin et al., 2010*; *Shi et al., 2012*; *van den Bogaart et al., 2010*). As a result, liposome–liposome fusion mediated by the four different SNARE complexes alone exhibit similar fusion rates (*Shen et al., 2007*; *Zwilling et al., 2007*; *Yu et al., 2013*). Taken together, the difference in CTD zippering energy contributes to the large variation in the fusion rate mediated by SNARE complexes and indicates a different number of SNARE complexes required for different fusion processes in vivo.

## Significance of the polarized energy distribution in membrane fusion

A common feature of our derived energy landscapes for all tested SNARE complexes is their polarized energy distribution, with much higher energy density at the C-terminus of CTD. This distribution allows SNAREs to increase their force output as CTD zippering draws two membranes into close proximity. Thus, as specialized engines for membrane fusion, SNAREs contain a built-in automatic transmission system that adjusts their force output to accommodate the large force change required for membrane fusion. This system ensures efficient and tight coupling between SNARE zippering and membrane fusion. A mismatch between the force output from SNAREs and the load from membranes would inevitably lead to dissipation of the SNARE zippering energy into heat, reducing the efficiency or the rate of membrane fusion. Thus, membrane fusion requires SNAREs to 'save the best for last'.

How does the SNARE complex focus its zippering energy to the C-terminus? *Li et al. (2014)* have recently shown that the association of the N-terminal half of VAMP2 to the N-terminal t-SNARE triggers folding of the t-SNARE C-terminal domain (*Figure 7A*, state ii). The ordered t-SNARE then serves as a template for fast and energetic VAMP2 zippering (*Gao et al., 2012*). Furthermore, tight association between v- and t-SNAREs near the C-terminus of CTD is achieved by key amino acids in that region, including the highly conserved phenylalanine residue shared by the v-SNAREs in all four SNARE complexes (*Figure 1—figure supplement 1*). Substitution of the phenylalanine residue with alanine abolishes the binary CTD transition in vitro (Ma L, Gao Y, Yang G, and Zhang YL, manuscript in preparation) and exocytosis (*Walter et al., 2010*). These observations strongly suggest that the polarized energy distribution of SNARE complexes is essential for membrane fusion.

In summary, as the molecular machine for membrane fusion, SNARE proteins share a working mechanism conserved from yeast to humans. They couple their step-wise folding/assembly to membrane fusion through a distinct half-zippered state. SNAREs contain a built-in transmission system that produces the highest forces at the very C-termini required for efficient membrane fusion. This unified mechanism provides a basis for dissecting the diverse functions of SNAREs in more detail.

# Materials and methods

## SNARE sequences, purification, and biotinylation

Amino acid sequences of the SNARE constructs used in our study are listed in *Figure 1—figure supplement 1*. The corresponding genes were codon-optimized, synthesized, subcloned into the protein expression pET-SUMO vector, and expressed in BL21(DE3) *Escherichia coli* cells as previously described (*Gao et al., 2012*). The proteins were purified using Ni-NTA resin (GE Healthcare Biosciences, Pittsburgh, PA) and biotinylated using biotin ligase (Avidity, Aurora, CO).

## High-resolution dual-trap optical tweezers

The tweezers were home-built and located in an acoustically isolated room with controlled temperature and air flow as previously described (*Moffitt et al., 2006*; *Sirinakis et al., 2011*). The machine was operated remotely through a computer interface written in LabVIEW (National Instruments, Austin, TX). The force and displacement measured by optical tweezers were calibrated by Brownian motion of polystyrene beads in optical traps before each single-molecule experiment. The beads were trapped in aqueous buffer in a microfluidic channel 0.2 mm in thickness, which was formed by sandwiching two coverslips with parafilm (*Zhang et al., 2012*).

## Single-molecule protein folding experiment

The cysteine-containing SNARE complex was reduced by TCEP or DTT, treated with dithiodipyridine (DTDP), mixed with the thiol-containing DNA handle in a typical 20:1 protein:DNA molar ratio, and cross-linked to the DNA handle overnight. An aliquot of the protein-DNA conjugate was mixed with anti-digoxigenin-coated beads and injected into the microfluidic channel. One DNA-bound bead was caught by one optical trap, brought close to a streptavidin-coated bead held in another optical trap, and formed a single SNARE-DNA tether. The SNARE complex was then pulled at a uniform trap separation speed or held at an approximately constant force or trap separation. The single-molecule folding experiment was performed at room temperature (22°C) in phosphate-buffered saline. An oxygen scavenging system was added to prevent photo-damage of the SNARE-DNA tether (*Gao et al., 2012*).

## Data analysis

Methods of data analysis are described in detail elsewhere (*Gao et al., 2012*; *Xi et al., 2012*) and are summarized here. The observed extension and energy changes contain contributions from the structured and unstructured parts of the SNARE protein as well as the DNA handle. The extension of the structured SNAREs was derived from the crystal structure of the SNARE complex and was assumed to be force-independent. The extensions of the unstructured polypeptide and the DNA handle were determined by the worm-like chain model (*Marko and Siggia, 1995*; *Smith et al., 1996*). Specifically, the extension (*x*) of a worm-like chain is related to the stretching force (*F*) and the contour length (*l*) by the Marko-Siggia formula

$$F(r) = \frac{k_B T}{P} \left[ \frac{1}{4(1-r)^2} + r - \frac{1}{4} \right], \tag{2}$$

where $r = x/l$, *P* is the persistence length of the polypeptide (0.6 nm) or DNA (30–50 nm), and $k_B T =$ 4.1 pN × nm the product of the Boltzmann constant and the room temperature. Extending a worm-like chain decreases its entropy. The associated energy increase can be obtained by integrating the force in *Equation 2* with respect to the extension, yielding

$$E(l,r) = \frac{k_B T}{P} \frac{l}{4(1-r)} (3r^2 - 2r^3). \tag{3}$$

For the two-state transitions of LD and CTD, we determined the unfolding probability, transition rates, and average state extensions and forces at different trap separations based on the measured extension and force trajectories using a two-state hidden Markov model. Then, we constructed a force-dependent energy landscape model that relates these experimental measurements to model parameters, including free energy of the folded state and transition state and their associated positions. Finally, we fit this model to the experimental data and determined the folding energy, folding energy barrier, and their associated structures.

The unfolding energy (ΔG) of a protein can be measured based on the mechanical work to reversibly unfold the protein, that is,

$$\Delta G = f_{1/2} \times \Delta X - E(\Delta l, \Delta x / \Delta l), \tag{4}$$

where $f_{1/2}$ is the measured equilibrium force, Δ*X* the corresponding extension change, and *E* the entropic energy of the unfolded polypeptide. Δ*x* and Δ*l* are the extension change and the contour length change, respectively, of the unfolded polypeptide associated with protein unfolding. However, neither Δ*x* nor Δ*l* in *Equation 4* is directly measurable and both are related to Δ*X* in a model-dependent manner (see Equations 12 and 13 in *Gao et al., 2012*). In addition, our experiments were not performed under exactly constant force, but constant trap separation for maximum spatiotemporal resolution (*Sirinakis et al., 2012*). As a result, a SNARE domain in a two-state transition experiences slightly different average forces in the folded state ($f_1$) and the unfolded state ($f_2$) (*Figure 2B*). Nevertheless, the average of the two state forces $f = (f_1 + f_2)/2$ remains constant, which we have simply referred to as force (*Gao et al., 2011*; *Figures 5 and 6*). Therefore, we constructed a detailed energy landscape model to quantitatively account for the correlation between protein structural transitions and the observed extension changes.

We chose the contour length of the unfolded polypeptide directly pulled by optical traps, which is 0.365 nm per amino acid, as a reaction coordinate to describe the extension change and the energy landscape associated with SNARE folding and unfolding. Different structural models were used for LD and CTD transitions: in LD transition, the two helices in $Q_a$ and R SNAREs fold and unfold symmetrically, whereas in CTD transition, R SNARE folds and unfolds along the pre-structured t-SNARE template (*Kummel et al., 2011*; *Gao et al., 2012*; *Li et al., 2014*). Using these structural models, we could fit the calculated extension to the measured FEC to determine the contour length parameter associated with each state. The total energy of the single-molecule system additionally includes the harmonic potential energy of two beads in optical traps. The folding energy of the structured part of the SNARE complex as a function of the contour length gives the folding energy landscape of SNARE folding. In our data analysis, this energy landscape was characterized by the free energy of the folded state and the transition state and their associated positions in the reaction coordinate, all relative to

the unfolded state. Both energy and positions were chosen as model parameters first to calculate the total system energy and the extension of the SNARE-DNA tether. Then these calculations were used to further compute the opening probability based on the Boltzmann distribution and the folding and unfolding rates based on Kramer's theory, as well as the extension change, for the transition of each SNARE domain. These values from model predictions were fit against the corresponding experimental data by the non-linear least-squares method, which yielded the best-fit model parameters, including the energy of the folded state and the transition state.

### Energy landscapes of trans-SNAREs

In our NTD structural model, we incorporated a detailed mechanism for t-SNARE folding induced by NTD association (*Li et al., 2014*). Because the kinetics of this coupled binding and folding process is unclear, we assumed that t-SNARE is gradually structured as VAMP2 starts to zipper from its N-terminus, and becomes fully structured when two-thirds of VAMP2 NTD has been zippered (*Figure 7D*). This forms a structure corresponding to the transition state of NTD association. Further zippering stabilizes NTD and forms the half-zippered state. Combined with the structural model for CTD transition, a complete model for assembly of the SNARE four-helix bundle was defined. This model also established the structure and the extension of the folded SNARE complex $h$ as a function of the contour length $l$. The membrane distance $d$ was determined by equating the SNARE pulling force to the membrane repulsive force, that is,

$$F(x) = -V'(d),\tag{5}$$

where $x = d - h - l^{3/5}p^{2/5}/2$ is the effective extension of the unfolded polypeptide with contour length $l$ and $V'$ the derivative of the membrane interaction energy (*Figure 7D*). The last term in the effective extension expression ($l^{3/5}p^{2/5}/2$) corrects for the residual extension of the unfolded polypeptide in the absence of external force when one end of the polypeptide is attached to the membrane, which is estimated to be half of the Flory radius of a semi-flexible chain (*Li et al., 2007*). Solving this non-linear equation at different contour lengths, we obtained the membrane distance $d$ at any SNARE zippering stage. The energy landscapes of trans-SNAREs are the total energy of SNAREs (including the folding energy and the elastic energy of unfolded polypeptide) and membrane interaction energy as a function of the membrane distance (*Figure 7B*). The calculations were performed using Matlab codes that are available as source codes.

## Acknowledgements

Aleksander A Rebane, the second author, performed exemplary work in accomplishing the complex data analyses herein reported. We thank Ying Gao for technical assistance and Jingshi Shen, Daniel Kummel, and Xinming Zhang for reading the manuscript. This work was supported by the NIH grants GM093341 to YZ and DK027044 to JER.

## Additional information

### Funding

| Funder | Grant reference number | Author |
| --- | --- | --- |
| National Institutes of Health | GM093341 | Yongli Zhang |
| National Institutes of Health | DK027044 | James E Rothman |

The funder had no role in study design, data collection and interpretation, or the decision to submit the work for publication.

### Author contributions

SZ, Conception and design, Acquisition of data, Analysis and interpretation of data, Drafting or revising the article; AAR, Analysis and interpretation of data, Drafting or revising the article; LM, Acquisition of data, Drafting or revising the article, Contributed unpublished essential data or reagents; GY, JC, Acquisition of data, Contributed unpublished essential data or reagents; MAM, Analysis and interpretation of data, Contributed unpublished essential data or reagents; FP, JER, YZ, Conception and design, Analysis and interpretation of data, Drafting or revising the article

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
