## [Decision Letter]

Thank you for sending your work entitled “SNARE assembly: common zippering intermediates and kinetics, different energetics” for consideration at *eLife*. Your article has been favorably evaluated by John Kuriyan (Senior editor), Axel Brunger (Reviewing editor), and three reviewers.

The Reviewing editor and the reviewers discussed their comments before we reached this decision, and the Reviewing editor has assembled the following comments to help you prepare a revised submission.

This manuscript reports single molecule optical trap force measurements for four different SNARE complexes, greatly extending previous work that was performed on the neuronal SNARE complex only. All four SNARE complexes show cooperative unfolding and refolding in the C-terminal half of the SNARE complex as well as an irreversible unfolding of the N-terminal half, but there are some differences in energetics. However, there are several major concerns:

1) The authors claim that the four SNARE complexes show common unfolding behavior, thus sharing a conserved zippering pathway. However, the unzipping behavior of the neuronal and endosomal SNARE complexes looks quite different from that of the GLUT4 and yeast SNARE complexes (shown in Figure 2 and Figure 2—figure supplement 1). After the C-terminal half unzipping, the neuronal and endosomal SNARE complexes require a 5 to 10 pN higher force to unzip the N-terminal half (the irreversible transition), whereas in the GLUT4 and yeast SNARE complexes, the N-terminal half unzipping occurs almost simultaneously with the C-terminal half unzipping. This point is important because the authors stress that the C-terminal half of the SNARE motif works as a sort of independent domain in SNARE-complex assembly for all four SNARE complexes. To make this scenario possible, it is presumed that the N-terminal half is stably assembled, while the C-terminal half then shows zippering in a selective manner after a certain pause. The authors need to characterize the force range in which they see the N-terminal half unzipping for each type of the SNARE complex (the number of molecules studied should be specified), and compare this force distribution with that used for the CTD folding and unfolding (e.g., shown in Figure 4). If these two force ranges (one for NTD unfolding and the other for CTD unfolding/refolding) are largely overlapping for a certain type of the SNARE complex, the SNARE-complex assembly may be essentially a one-step process in an all-or-none manner for that type of the SNARE complex.

2) All four SNARE complexes show a large hysteresis in their unfolding and refolding behavior. Thus, there is no guarantee that the N-terminal half unzipping and rezipping occurs at a same force level. Ideally, to confirm that the two-step model applies to the assembly process as well, re-assembly experiments should be performed. At the minimum, this limitation of the current experiments needs to be discussed.

3) The authors argue that the average force generated during one-third of CTD folding is 8 pN and zippering of the last one-third produces an average force up to 31 pN. Presumably, the authors reached these force values by taking derivatives of the energy landscape of the two-state model (in the zero-force case) that was also used to derive the folding energy value (if this is not the case, please provide the details of estimating these force values). An estimation of the folding energy value and position of the energy barrier seems reasonable, but the calculation of detailed force values is questionable using this simple two-state model. To derive the force value, one needs to know the local curvature of the energy landscape as demonstrated in Woodside et al. Science (2006) 314, 1001. Actually, it seems that the estimated force values do not well match with the experimental observation. For instance, the CTD of the neuronal SNARE complex shows active unfolding with 16 pN that is only half of the 31 pN force.

4) Statistical information is missing throughout the manuscript. For example, in Figure 3 and Figure 4, how many independent SNARE complexes are studied for each data?

5) A concern is the effect of the spacers linking the different proteins into the chimeric constructs. The authors should provide evidence that the chimera likely zipper into the correct structure. More importantly, they should show that the zippering kinetics and/or pathway are unaffected. For example, the authors could investigate the effect of different spacer lengths or topologies on zippering kinetics.

6) The manuscript is very long and difficult read. There are many instances of incorrect English usage and/or confusing wording. While there is there is no formal word limit for this journal, it is not clear that the current length is warranted given the data presented. On the other hand, there are many instances where a more concise explanation would have improved the readability, and other instances where more details in the analysis would have been helpful. Such details should be provided in the Methods section. The authors are strongly encouraged to revise their text extensively to appeal to the broad readership of *eLife*.

---

## [Author Response]

*1) The authors claim that the four SNARE complexes show common unfolding behavior, thus sharing a conserved zippering pathway. However, the unzipping behavior of the neuronal and endosomal SNARE complexes looks quite different from that of the GLUT4 and yeast SNARE complexes (shown in*
Figure 2
*and*
Figure 2—figure supplement 1*). After the C-terminal half unzipping, the neuronal and endosomal SNARE complexes require a 5 to 10 pN higher force to unzip the N-terminal half (the irreversible transition), whereas in the GLUT4 and yeast SNARE complexes, the N-terminal half unzipping occurs almost simultaneously with the C-terminal half unzipping. This point is important because the authors stress that the C-terminal half of the SNARE motif works as a sort of independent domain in SNARE-complex assembly for all four SNARE complexes. To make this scenario possible, it is presumed that the N-terminal half is stably assembled, while the C-terminal half then shows zippering in a selective manner after a certain pause. The authors need to characterize the force range in which they see the N-terminal half unzipping for each type of the SNARE complex (the number of molecules studied should be specified), and compare this force distribution with that used for the CTD folding and unfolding (e.g., shown in*
Figure 4*). If these two force ranges (one for NTD unfolding and the other for CTD unfolding/refolding) are largely overlapping for a certain type of the SNARE complex, the SNARE-complex assembly may be essentially a one-step process in an all-or-none manner for that type of the SNARE complex*.

We agree with the reviewers that distinct distributions of the CTD equilibrium force and the NTD unzipping force will help reinforce our conclusion that NTD and CTD assemble and disassemble in kinetically distinct steps. Following the reviewers’ suggestion, we have prepared the histogram distributions of the CTD equilibrium force and the NTD unzipping force for all four SNARE complexes and shown in a new Figure 2—figure supplement 2. Neuronal, yeast, and endosomal complexes exhibit mainly non-overlapping distributions, confirming distinct steps of CTD and NTD assembly.

GLUT4 SNARE complex has overlapping CTD and NTD force distributions. However, this overlap does not contradict the stage-wise assembly and disassembly of the GLUT4 SNARE complex. Strictly speaking, the CTD equilibrium force and the NTD unzipping force are different quantities and are not directly comparable. The equilibrium force is defined as the force under which the CTD has half unfolding probability (Figure 3), whereas the NTD unzipping force is defined as the force that NTD first unzips. At a low pulling or force-loading speed, NTD may unzip before the pulling force reaches the CTD equilibrium force (see Figure 2—figure supplement 1 for an example). However, in this case, CTD typically has already folded and unfolded for more than 30 times. If we compare the force and time difference at which CTD and NTD first unzip for each SNARE complex, the CTD always unzips earlier and at lower force than NTD for all four SNARE complexes (Figure 2—figure supplement 2). Therefore, the CTD and the NTD unzips in a sequential manner. This observation cannot be explained by an “all-or-none” assembly mechanism, because such a two-state process also implies a simultaneous and one-step unfolding of the whole SNARE bundle, which is not observed.

It must be pointed out that the differences between average CTD and NTD unzipping forces depend on the pulling speed, because the NTD unzipping process is not in equilibrium (5). A lower pulling speed will decrease the force difference, mainly because the average NTD unzipping force decreases. Under a constant force, distinct CTD and NTD assembly or disassembly can still be revealed by their different lifetimes. The average lifetime of the folded CTD is generally less than 50 milliseconds under our experimental conditions (Figure 3), whereas the average lifetime of the folded NTD is typically greater than 10 s in these experiments. Without a long NTD lifetime, it would be impossible for us to collect more than tens of thousands of CTD transition events at constant forces required to accurately determine the CTD opening probability and transition rates shown in Figure 3. Thus, a non-overlapping force distribution is a sufficient, but not necessary condition for distinct NTD and CTD assembly and disassembly (consider the fact that all protein complexes dissociate even at zero force given sufficient time).

The lifetime difference between CTD and NTD becomes even greater for trans-SNARE complexes bridging two membranes. In this case, CTD zippering experiences a much greater opposing force than NTD zippering (Figure 5). As a result, the force-dependent half-zippered SNARE state is greatly stabilized by the apposed membranes.

In conclusion, our data are consistent with kinetically distinct NTD and CTD assembly and disassembly, but not simultaneous and one-step SNARE assembly. However, our data do not rule out cooperative interactions between NTD and CTD assembly. Instead, NTD assembly facilitates CTD assembly by contributing a coupling energy estimated to be 6 kT, as shown in our previous work (12).

In the revised manuscript, we have therefore added two supplemental figures, Figure 2—figure supplement 1 and Figure 2—figure supplement 2. These figures are referred to in the following sentences:

“The FECs show that all four SNARE complexes sequentially unfolded via two reversible transitions and one or two irreversible unfolding steps (Figure 2 and Figure 2—figure supplement 1 and Figure 2—figure supplement 2).”

*2) All four SNARE complexes show a large hysteresis in their unfolding and refolding behavior. Thus, there is no guarantee that the N-terminal half unzipping and rezipping occurs at a same force level. Ideally, to confirm that the two-step model applies to the assembly process as well, re-assembly experiments should be performed. At the minimum, this limitation of the current experiments needs to be discussed*.

We agree with the reviewers that it is technically difficult to reversibly unzip and re-zip the N-terminal half of the SNARE complex under constant forces. We once held neuronal SNARE complex at a constant force upon CTD unfolding for an overall 30 min and saw no re-folding, indicating a large energy barrier for NTD refolding as well as unfolding at high forces (>12 pN). Consequently, we had to relax the complex to observe NTD refolding at a lower force (1-8 pN, see Figure 2—figure supplement 3, 5–7 and Figure in (12)). This slow NTD association justifies the existence of the partially zippered neuronal SNARE complex, because de novo or all-or-none SNARE assembly will be too slow to match the speed (<1 ms) required for the fast neurotransmission.

The SNARE reassembly processes observed in our experiments under low forces generally do not show any intermediates, because the lifetime of the half-zippered state is too short to be detected by our machine as is expected. As we reported before (12) and modeled in Figure 5, the half-zippered state is force-dependent and stabilized by the short-ranged membrane opposing force. The state is stabilized, because its CTD zippering is opposed by large membrane repulsive force whereas its NTD association only meets small opposing force. Such a large force gradient cannot be mimicked by the linear force provided by optical tweezers to stabilize the half-zippered state as efficiently as the opposing membranes at the lower forces.

Finally, by crosslinking neuronal SNARE complex at a strategic position, we have recently been able to detect distinct reversible transitions in both NTD and CTD, confirming the stage-wise SNARE assembly and disassembly (Ma, L., Gao, Y., Yang, G., and Zhang, Y. L., manuscript in preparation).

To address the reviewers’ concern, we de-emphasized “assembly” by changing it to “assembly and disassembly”. In addition, following the reviewers’ suggestion and our above considerations, we added the following sentences:

“Correspondingly, the metastable half-zippered SNARE complex in the absence of a force load was stabilized by the short-ranged membrane opposing force (Figure 5) together with regulatory proteins, such as complexin (47; 12; 31). In contrast, the optical trapping force used in our experiments was long-range, with a typical force constant of 0.1 pN/nm, compared to an average force constant of 7.1 pN/nm for the membrane force opposing CTD assembly. As a result, the half-zippered SNARE state was typically short-lived (<0.2 ms) upon reassembly of the SNARE complex at the low forces favoring NTD association (Figure 2—figure supplement 3), because the trapping force opposing CTD zippering remained small immediately after NTD was zippered. Thus, the membrane’s repulsive force was an integral component of SNARE assembly and regulation.”

*3) The authors argue that the average force generated during one-third of CTD folding is 8 pN and zippering of the last one-third produces an average force up to 31 pN. Presumably, the authors reached these force values by taking derivatives of the energy landscape of the two-state model (in the zero-force case) that was also used to derive the folding energy value (if this is not the case, please provide the details of estimating these force values). An estimation of the folding energy value and position of the energy barrier seems reasonable, but the calculation of detailed force values is questionable using this simple two-state model. To derive the force value, one needs to know the local curvature of the energy landscape as demonstrated in Woodside et al. Science (2006) 314, 1001. Actually, it seems that the estimated force values do not well match with the experimental observation. For instance, the CTD of the neuronal SNARE complex shows active unfolding with 16 pN that is only half of the 31 pN force*.

Yes, we estimated the CTD zippering force based on a simplified energy landscape model for a two-state process. We have added Figure 5–figure supplement 2 to clarify and justify our estimation. Please see this figure as well as its legend for our replies to the reviewers’ comments.

*4) Statistical information is missing throughout the manuscript. For example, in*
Figure 3
*and*
Figure 4*, how many independent SNARE complexes are studied for each data*?

To address this point, we added the required statistical information in the legend of Table 1 and in Figure 2—figure supplement 2, including the number of transitions and single molecules scored. We have also added more statistical information when average values are given. Specifically, in the table legend, we added the following sentences in the main text:

“The equilibrium force distribution, the number of transitions and the number of single molecules scored for CTD transitions are shown in Figure 2—figure supplement 2. For parameters related to LD transitions, a total of 18, 35, 11, and 24 LD transitions in single neuronal, GLUT4, endosomal, and yeast SNARE complexes were scored, respectively.”

“Specifically, neuronal, GLUT4, endosomal, and yeast SNARE complexes in the half-zippered state had their R SNAREs unzipped to −1, +3, +1, and +3 amino acids relative to the ionic layer, respectively, whereby the positive sign designates the C-terminal amino acids. The standard deviation of all positions was less than 3 amino acids (Table 1).”

“Nonlinear least-squares fitting of the model matched the experimental data well (Figure 3), which revealed the free energy of the folded state and the transition state and their relative positions (Table 1). The CTD folding energy of neuronal, GLUT4, endosomal, and yeast SNARE complexes were −27 (±5, s.d. throughout the text) k_B_T, −23 (±4) k_B_T, −16 (±2) k_B_T, and −13 (±3) k_B_T, respectively.”

*5) A concern is the effect of the spacers linking the different proteins into the chimeric constructs. The authors should provide evidence that the chimera likely zipper into the correct structure. More importantly, they should show that the zippering kinetics and/or pathway are unaffected. For example, the authors could investigate the effect of different spacer lengths or topologies on zippering kinetics*.

Several lines of evidence, from both our experiments and theoretical calculations, show that the effect of the spacers on SNARE zippering energy and kinetics is minor and can be accounted for quantitatively. First, results from the CD measurement and gel filtration are consistent with a fully folded four-helix bundle (Figure 1—figure supplement 2 and Figure 1—figure supplement 3); Second, the folding energy and kinetics of LD and CTD in neuronal SNARE complex match those of our earlier measurements based on a different SNARE construct without any long spacers used. In particular, the spacer sequence has minimal effort on CTD transition (Figure 3), because the extension of the spacers does not change in this process. Third, the spacer sequence introduced between syntaxin and SNAP-25 does not significantly affect the function of the t-SNARE to mediate membrane fusion in vitro (Figure 1—figure supplement 4). Fourth, results from the single-molecule manipulation experiments are generally homogenous among different molecules pulled and consistent with correctly folded four-helix bundle structures. If alternative folded conformations had occurred in our experiments, they could have been identified from their different assembly/disassembly kinetics (Figure 2—figure supplement 3). Fifth, for yeast SNARE complex, the derived LD and CTD transitions are corroborated by the experimental results from truncated SNARE complexes (Figure 2—figure supplement 4). Finally, the minor effect of the spacer sequences on SNARE assembly is confirmed by our theoretical calculations (Figure 4–figure supplement 2). The spacer sequence between t and v-SNAREs is expected to slightly destabilize NTD by ∼4.3 k_B_T due to stretching of the spacer, compared to ∼30 k_B_T folding energy of the NTD (12). Because this destabilized NTD still has a much larger lifetime than CTD, our conclusion regarding distinct kinetics of NTD and CTD assembly is justified.

To address the reviewers’ concerns, in our revised manuscript, we have clarified the minor effect of the spacer sequences on SNARE transition energy and kinetics (Figure 4-figure supplement 2). In addition, we have emphasized this point in the main text as follows:

“Furthermore, the chimeric neuronal SNARE complex reveals folding energy and kinetics (see below) consistent with our recent reports based on a different SNARE construct in which syntaxin and VAMP2 were crosslinked at their N-termini by a disulfide bond (12).”

“The CTD folding energy and the equilibrium rate (∼100 s^−1^) of the neuronal SNARE complex were very close to the energy (28 ± 3 k_B_T) and the rate (∼160 s^−1^) reported earlier (12), indicating that the spacer sequences in the chimeric construct used here have minimal effect on the folding energy and kinetics of the SNARE complex.”

“Correcting for the minor effect of the spacer sequence added between syntaxin and SNAP-25 (Figure 4–figure supplement 2), we obtained LD zippering energy of −10 (±2) k_B_T for neuronal SNARE complex, consistent with our previous measurement of −8 (±2) k_B_T.”

Based on above extensive experimental evidence and theoretical calculations, we think our major conclusions are justified, and not affected by the chimeric SNARE constructs used in this work. Thus, we did not pursue studies to vary the length of the spacer sequence.

*6) The manuscript is very long and difficult read. There are many instances of incorrect English usage and/or confusing wording. While there is there is no formal word limit for this journal, it is not clear that the current length is warranted given the data presented. On the other hand, there are many instances where a more concise explanation would have improved the readability, and other instances where more details in the analysis would have been helpful. Such details should be provided in the Methods section. The authors are strongly encouraged to revise their text extensively to appeal to the broad readership of eLife*.

Following the reviewers’ suggestion, we have shortened the main text and made revisions to add the necessary detail to the “Data Analysis” section of the Materials and Methods. The revised text has also been professionally edited.